# Cadmium-Associated Molecular Signatures in Cancer Cell Models

**DOI:** 10.3390/cancers13112823

**Published:** 2021-06-05

**Authors:** Claudio Luparello

**Affiliations:** Dipartimento di Scienze e Tecnologie Biologiche, Chimiche e Farmaceutiche (STEBICEF), Università di Palermo, 90128 Palermo, Italy; claudio.luparello@unipa.it

**Keywords:** cadmium, differential expression, gene signature, in vitro cell models, breast cancer, gastric cancer, colon cancer, liver cancer, lung cancer, nasopharyngeal cancer

## Abstract

**Simple Summary:**

The exposure of cancer cells to cadmium compounds may be associated with the acceleration of tumor progression. It is known that cadmium is a transcriptional regulator, and the study of differentially expressed genes has enabled the identification and classification of cadmium-associated molecular signatures as useful biomarkers that are potentially transferable to clinical research. This review recapitulates the studies that report the detection of such signatures in breast, gastric, colon, liver, lung, and nasopharyngeal tumor cell models, as specifically demonstrated by individual gene or whole genome expression profiling.

**Abstract:**

The exposure of cancer cells to cadmium and its compounds is often associated with the development of more malignant phenotypes, thereby contributing to the acceleration of tumor progression. It is known that cadmium is a transcriptional regulator that induces molecular reprogramming, and therefore the study of differentially expressed genes has enabled the identification and classification of molecular signatures inherent in human neoplastic cells upon cadmium exposure as useful biomarkers that are potentially transferable to clinical research. This review recapitulates selected studies that report the detection of cadmium-associated signatures in breast, gastric, colon, liver, lung, and nasopharyngeal tumor cell models, as specifically demonstrated by individual gene or whole genome expression profiling. Where available, the molecular, biochemical, and/or physiological aspects associated with the targeted gene activation or silencing in the discussed cell models are also outlined.

## 1. Introduction: A Short Excursus on Cadmium and Eukaryotic Cell Systems

Cadmium (Cd) is an underground mineral extracted as part of zinc deposits which, along with its compounds, exhibits a broad range of applications in the industry spanning from battery components to stabilizers for plastics, electroplated coatings for non-ferrous metals and dye pigments. Volcanic eruption and the erosion of Cd-containing rocks represent the major natural sources of the metal; on the other hand, humans are mainly exposed to Cd via cigarette smoking and, to a lesser extent, food ingestion, inhalation of polluted air, and the ingestion of contaminated drinking water [1].

Cd is not essential for the human body and does not exert any useful cellular metabolic effect. The cellular uptake of free Cd ions and complexes of Cd with small organic molecules involves various ion channels, carriers, and ATPase pumps, whereas Cd-protein complexes are internalized via receptor-mediated endocytosis, as extensively reviewed by the authors in [2]. Once accumulated in the intracellular milieu, the metal may interact with the thiol groups of cysteines present in cellular proteins, thus inducing an impairment of the functions of the enzymes located in different cellular compartments. The numerous biological targets and complex mechanisms of the action of Cd ions have been explored in depth in previously-published reviews [3,4,5]. It is known that Cd can cause mitochondrial dysfunction, exert genotoxic and epigenotoxic effects, and can also interfere with a number of cell proliferation-linked signalization pathways. Reports in the literature have shown that mitochondria are one of the main intracellular targets for Cd toxicity due to the blocking of the electron transfer chain at complex III, which is considered the principal site for the production of reactive oxygen species (ROS), thereby favoring the over-production of the latter at the expense of ATP. This leads to a dissipation of mitochondrial transmembrane potential with a consequent alteration of not only the cell’s redox status but also mitochondrial and nuclear gene expression and the genome integrity [6,7]. In fact, it is widely acknowledged that the genome-related effects of Cd are based upon its ability to damage the antioxidant defense and DNA repair systems, and also due to the replacement of zinc by Cd in p53 with a consequent impairment of protein activity, thereby triggering the occurrence of DNA strand breaks and chromosomal aberrations [8,9]. The inhibition of p53, coupled with the direct stimulation of mitogenic signals via calcium and inositol triphosphate second messengers, is also responsible for Cd-dependent enhanced cell proliferation. On the other hand, in specific cell systems the metal was conversely found to induce endoplasmic reticulum stress due to severe ROS and calcium signaling, which ultimately leads to apoptotic and/or autophagic cell death [10,11,12,13]. As a further DNA-directed activity, evidence has been produced that Cd may be regarded as an epigenetic modulator, given its ability to alter the global DNA methylation pattern via the down-regulation of DNA methyltransferase activity and demethylation which, upon prolonged exposure, turns into enhanced enzyme activity and genome hypermethylation. These opposite alterations may result in the up-regulation of cellular protooncogenes and the silencing of oncosuppressor genes, respectively [14,15].

Evidence collected in the literature has highlighted that the exposure of cancer cells to Cd compounds may be associated with the development of more malignant phenotypes, thereby contributing not only to the onset of cell transformation in the tumor initiation stage, which has been widely reviewed in several articles e.g., in [16,17], but also to the acceleration of tumor progression. An example of this aspect is the growth-promoting “metalloestrogen” role played by Cd on estrogen receptor α-positive breast cancer cells. On the other hand, based upon the intrinsic cytospecificity, Cd has been proven to act as a two-edged sword, conversely down-regulating viability and causing the death of a number of tumor cell lines, such as those derived from triple-negative breast cancer and hepatocarcinoma [18,19,20]. Cd is a transcriptional regulator that induces molecular reprogramming [21]; therefore, the study of differentially expressed genes has enabled the identification and classification of the molecular signatures inherent in human neoplastic cells upon Cd exposure as useful biomarkers that are potentially transferable to clinical research. The goal of this review is to recapitulate selected studies that report the detection of such signatures, as specifically demonstrated by individual gene or whole genome expression profiling, in particular focusing on breast, gastric, colon, liver, lung and nasopharyngeal tumor cell models. Where available, the molecular, biochemical and/or physiological aspects associated with the targeted gene activation or silencing in the discussed cell models will be also briefly outlined.

## 2. Molecular Signatures in Breast Cancer Cells

The search for Cd exposure-associated molecular signatures has been conducted in two cell lines, i.e., MDA-MB231 and MCF-7. The former was established from a pleural effusion of a triple-negative breast cancer (TNBC) of basal morphology, which was negative for estrogen receptor (ER) α and progesterone receptor (PR) expression and its p53 was inactivated by a mutation in codon 280 of exon 8. The ERα-positive and PR-negative MCF-7 cell line was established from a pleural effusion of a highly hormone-responsive malignant adenocarcinoma [22,23,24].

### 2.1. Molecular Signatures in MDA-MB231 Cells

Concerning TNBC cells, about a decade ago papers published by my group of researchers first contributed to this issue by investigating the effect of administering different concentrations of CdCl_2_ to MDA-MB231 cells and also comparing the tumor cell behavior with that of HB2 cells, a nonneoplastic immortalized line from the human breast epithelium [25]. The obtained results demonstrated the cytotoxic effect of the molecule with a 50% inhibitory concentration (IC_50_) at 96 h of 5 μM, which conversely was ineffective in modifying the survival and growth of HB2 cells [18]. Among the biological aspects of MDA-MB231 cells studied under this experimental condition, molecular signatures were searched that demonstrated an up- or down-regulation of the expression levels of genes coding for heat shock proteins (hsp), metallothioneins (MT), cytochrome oxidase subunits, and other factors related to apoptosis, signal transduction, and growth control, as summarized sinoptically in Table 1 (see refs. [18,26,27,28,29,30]).

In light of the literature on the subject, some of the produced data allow for the following comments to be made.

#### 2.1.1. Expression Levels of MTs

MTs are a group of cysteine-rich, low molecular weight proteins able to bind metals and act as controllers of cellular homeostasis through protection against oxidative stress, heavy metal toxicity, and DNA damages. The exposure of MDA-MB231 cells to CdCl_2_ resulted in the up-regulation of *MTIA* and *MTIF* whilst *MTIG* was down-regulated. It is noteworthy that MTIA mRNA and proteins were proven to be induced by the exposure of human proximal tubule cells to Cd concentrations, eliciting a downfall of cell viability [31]. In addition, *MTIA* overexpression provided HEK293 embryonic kidney cells with more resistance to the Cd administration than other MT isoforms [32]. Taking the data from the literature into account, it is conceivable that the up-regulation of this MT isoform in the MDA-MB231 cells may represent a cellular defense reaction specifically set up against Cd intoxication. Interestingly, Chang et al. [33] validated MTIA mRNA expression as a molecular marker for renal dysfunction in occupational Cd exposure. Concerning the other two MTs, Albrecht et al. [34] reported a significant increase in the MT1F transcripts in RWPE-1 human prostate epithelial cells at the time points immediately preceding cell death due to exposure to 3, 6 and 12 mM Cd^2+^ over a 13-day period, which may be consistent with the cytotoxic effect exerted on MDA-MB231 cells. Moreover, the down-regulation of *MTIG*, whose reduction was found to induce glutathione depletion and lipid peroxidation in hepatocellular carcinoma cells [35], may be a contributing factor to MDA-MB231 cell death.

#### 2.1.2. Expression Levels of Heat Shock Proteins

The up-regulation of *HSPB1* and the down-regulation of *HSPA5*, *HSPD1*, *HSP90AB1* and *TRAP1* in CdCl_2_-treated MDA-MB231 cells has been demonstrated. It is known that oxidative damage and ROS generation may selectively trigger a significant decrease in the *HSPA5* levels [36] and therefore these data are consistent with the findings of a study into the increase in mitochondrial respiratory activity and the accumulation of ROS in treated MDA-MB231 cells, as shown by Cannino et al. [27]. Moreover, the down-regulation of *HSPD1* and *TRAP1*, coding for two mitochondrial chaperonins, could be responsible for the accumulation of ROS as reported by the authors in [37,38], whereas the down-regulation of *HSPA8* may contribute to the decrease in cell protection against ROS-mediated lipid peroxidation during the oxidative challenge. Interestingly, Masuda et al. [39] suggested that Bcl-2 may be one of the regulators of *TRAP1* expression, which is consistent with the data demonstrating the concomitant down-regulation of *BCL2* and *TRAP1* in exposed MDA-MB231 cells. Exposure to 10 mM CdCl_2_ for 24 h was found to be responsible for the decrease in the expression level of *HSP90AB1*, contributing to the metal-promoted lethal effect on cholinergic neurons [40]. Kindas-Mugge et al. [41] demonstrated the decrease in the proliferation rate of *HSBP1*-overexpressing MDA-MB231 cells. It is conceivable that the restraining effect of hsp27, the product of *HSBP1*, on cell growth behavior could be jointly responsible with the induction of cell death for the halving of the MDA-MB231 cell numbers observed after 96 h-treatment with 5 µM CdCl_2_.

#### 2.1.3. Expression Levels of Cytochrome Oxidase Components

The exposure of MDA-MB231 cells to CdCl_2_ resulted in the down-regulation of *COX2* and *COX4*, thereby suggesting a relationship between the reduction in the amount of COX and the production of ROS, given that mitochondrial proteins are some of the earliest cellular targets because of their immediate proximity to the ROS-generating sources [27].

#### 2.1.4. Expression Levels of Proliferation- and Apoptosis-Related Genes

The exposure of MDA-MB231 cells to CdCl_2_ determined the down-regulation of *BCL2* and *AEG1* and the up-regulation of *DAPK*, *WAF1*, *RIPK*, *PLP2*, *FOS*, *JUN*, and several genes coding for caspases, the enzymatic components of the apoptotic machinery. This result suggests that the metal directs the tumor cells towards a type of death which shares several features with programmed cell death. In fact, the Bcl-2 protein is an apoptosis-suppressor factor, whereas death-associated protein kinase-1 is involved in the onset of the death pathways of apoptosis, as well as autophagy and programmed necrosis. In MDA-MB231 cells, cyclin-dependent kinase inhibitor 1 was proven to negatively modulate the cell cycle progression due to binding to both the cyclin–CDK complex and proliferating the cell nuclear antigen, and the *RIPK1* product, a kinase forming part of the tumor necrosis factor receptor-1 (TNFR1) complex I, which is known to be one of the most important upstream mediators of NF-κB signaling as well as an important regulator of cell death [42,43,44]. In breast cancer, the astrocyte elevated gene-1 protein was found to be an agonist of the Wnt pathway which regulates cell proliferation and is involved in the control of the NF-kB pathway and the expression of proliferation-promoting *FOS*/*JUN* genes [29,45]. The knockdown of endogenous *AEG1* was proven to sensitize MDA-MB-231 cells to TRAIL-induced apoptosis both in vitro and in vivo. Therefore, in line with the data of Zhang et al. [46], the possibility that CdCl_2_-mediated *AEG1* down-regulation may facilitate the intrinsic and extrinsic apoptosis pathways via the decrease in *BCL2* expression levels and *CASP8* up-regulation could be considered. In addition, *PLP2* over-expression was also found to be correlated with the increase in *CASP8* expression levels [30].

#### 2.1.5. Expression Levels of MAPKs

CdCl_2_ treatment on MDA-MB231 cells was efficient in decreasing the expression levels of *MAPK11* and −*14* and increasing that of *MAPK12*, although an opposite variation in the accumulation of the protein product of the latter, i.e., mitogen-activated protein kinase p38γ was recorded. Zhong et al. [47] demonstrated that in case of oxidative stress, the activation of the MAPK11/12/13/14/NF-kB pathway may induce protective autophagy via the transcriptional activation of the autophagy-related genes *MAP1LC3B*, *BAG3*, and *HSPA1A* in HeLa cells treated with an anticancer copper complex. Therefore, it might be assumed that MAPK down-regulation following the exposure of breast cancer cells to CdCl_2_ may contribute to the cytotoxic effect through the inhibition of such a protective cell response. 

More recently, additional Cd-related molecular signatures in MDA-MB231 cells have been brought to light. Wei et al. [48,49] reported that non-cytotoxic concentrations of CdCl_2_ (0.5–1 µM) trigger cell migration, an ability which in vivo can result in an enhanced metastatic attitude during TNBC progression. Using luciferase reporter assays, after prolonged Cd treatment (3 µM for 4 weeks), they demonstrated the up-regulation of the transcriptional activity of the T-cell factor/lymphoid enhancer-binding factor (TCF/LEF), whose contribution to the Wnt signaling machinery is well-known, and Snail, a target of the Notch1 pathway primarily involved in epithelial-mesenchymal transition (EMT) processes as demonstrated by knockdown experiments [50,51]. The effect on the TCF/LEF appeared to be due to Cd-induced signaling via the interaction with the integrin/focal adhesion kinase (FAK) pathway leading to the inhibition of GSK3β and consequent β-catenin accumulation and nuclear translocation. On the other hand, Cd induced the elevation of Snail transcription by up-regulating the activity of its promoter. Moreover, their results also suggested that Cd may inhibit miR-200 that post-transcriptionally restrains the translation of ZEB-1 which, in turn, is a direct suppressor of the transcription of the gene encoding the epithelial marker E-cadherin, thus further contributing to the regulation of EMT [52].

### 2.2. Molecular Signatures in MCF-7 Cells

Concerning the ERα-positive MCF-7 cell line, in 2013 Lubovac-Pilav et al. [53] investigated the effect of chronic (up to 6 months) 10^−7^ M CdCl_2_ exposure on the gene expression pattern by high throughput microarray technology followed by hierarchical clustering analysis and functional annotations. More recently, Liang et al. [54] complemented the analysis by producing data on the epigenomic as well as the transcriptomic profiles induced by short-term (72 h) 60 µM CdCl_2_ treatment, further identifying also in this case the critical pathways and genes via bioinformatic studies. Their cumulative results provide a broad picture of the impact of the exposure of breast cancer cells to the metal compound at the gene signature level, with 795 differentially expressed genes identified by the authors in [53] and 997 differentially expressed genes identified by the authors in [54]. In particular, taking their most significant findings on the highest changes in gene expression levels and the data from the literature into account, the following comments can be made. 

#### 2.2.1. Expression Levels of Breast Cancer-Associated Factors

Ninety-seven genes identified by the authors in [53] were known to be breast cancer-associated, thereby confirming that Cd is active in accelerating breast carcinogenesis by modulating the expression of genes involved in tumor progression. Table 2 shows those breast carcinogenesis-associated genes positioned among the top 30 up- and down-regulated ones upon chronic metal exposure, all of which were also linked to endocrine disruption effects, while some of them were also linked to the regulation of cell proliferation and differentiation (e.g., *CRABP1* and *CCNE1*) and ROS production (e.g., *UCP2*). The identification of gene signatures that were significantly regulated by short-term CdCl_2_ exposure and strongly associated to breast carcinogenesis through the protein-protein network analysis led Liang et al. [54] to extract *TXNRD1* and *CCT3* from the list of 400 validated genes as those endowed with the highest degree of expression change, hazard ratio difference, and degree of connectivity. The first gene codes for thioredoxin reductase 1, an intracellular redox sensor and antioxidant enzyme whose deregulation may be involved in breast cancer initiation [55]. *CCT3* protein product is the chaperonin containing TCP1 subunit 3, a component of a complex which catalyses the correct folding of proteins involved in cytoskeletal assembly and the cell cycle and whose involvement in the promotion of breast cancer cell proliferation and metastasis via Wnt has been recently acknowledged [56,57]. It is also of note that one of the most significant Gene Ontology (GO) terms in which a large amount of the genes validated by Liang et al. [54] were enriched was the “Wnt signaling pathway”, which is known to be highly dysregulated in breast tumor [58].

#### 2.2.2. Expression Levels of Metal Ion- and Xenobiotic-Binding Factors

The chronic exposure of MCF-7 cells to CdCl_2_ was proven to affect the expression level of the genes coding for not only Cd- but also calcium-, zinc- and iron-binding proteins due to the mimesis and interference played by Cd towards the other metals. Table 3 shows those genes positioned among the top 30 up- and down-regulated ones in the “GO molecular functions: metal ions” category. More specifically, one of them was subcategorized as “cadmium ion binding” (*MT1F*), five were subcategorized as “calcium ion binding” (*ANXA3*, *ANXA2P2*, *ATP2A3*, *FKBP9* and *OCM*), one was subcategorized as both (*MT1X*), and four were subcategorized as “zinc ion binding” (*CBFA2T3*, *MT2A*, *PDLIM1* and *ZNHIT2*). This was consistent with the ability of Cd to mimic other divalent cations thus stimulating the expression of metal transporters. The up-regulation of *MT1* expression after the short-term exposure of MCF-7 cells to Cd was also reported by Darwish et al. [59] who also demonstrated the concurrent over-expression of *MDR1* and *MRP2* genes, coding for the xenobiotic transporters multidrug resistance-associated protein 1 and 2, respectively, which are known to also be involved in Cd efflux as a cytoprotective reaction to cell intoxication [60].

#### 2.2.3. Expression Levels of Cell Growth-Associated Factors

Another GO category which was analyzed in chronically CdCl_2_-intoxicated MCF-7 cells was “GO biological process: cell growth” [54]. Two were the genes found among the top 30 up-regulated ones, that is, cyclin E1-coding *CCNE1* which was subcategorized under the GO terms “cell division” and “cell cycle”, and the zinc ion transporter-coding *CBFA2T3* included in the “cell proliferation” subcategory. It is known that cyclin E1 is an oncogene which controls the G_1_/S phase transition and its expression is directly related to the aggressive potential of breast cancer [61]; on the other hand, the association of metal treatment on breast tumor cells and *CBFA2T3* over-expression is still undetermined. In addition, consistent with the Cd-induced positive effect on breast tumor cell growth, Siewit et al. [62] demonstrated the up-regulation of *CCND1* and *MYC*, coding for cyclin D1 and c-myc proteins, and the down-regulation of *CDKN1A*, coding for the p21/WAF1 protein, after 24 h of MCF-7 cell exposure to micromolar CdCl_2_ concentrations.

#### 2.2.4. Expression Levels of Methyltransferases

Following MCF-7 cell treatment with 0.1, 0.5 and 1 **m**M CdCl**_2_** for 24 h and 48 h, Ghosh et al. [63] reported the Cd dose-dependent increase in the expression of the *PRMT5* and *EZH2* genes coding for the proteins arginine methyltransferase 5 and histone-lysine N-methyltransferase enhancer of zeste homolog 2, respectively, which are two critical epigenetic modulators responsible for the development of multiple cancer histotypes. The experimental data collected suggested that such expressional modulations could be due to CpG island demethylation in the gene promoter sites as a positive feedback loop of the Cd-induced down-regulation of methyltransferase activity, and/or the Cd-promoted increase in the levels of NFYA and E2F1 transcription factors and their subsequent enriched recruitment to the demethylated gene promoters.

#### 2.2.5. Expression Levels of Heat Shock Proteins

The dose-dependent up-regulation of the *HSPA2* gene, coding for hsp70 protein, was reported by Darwish et al. [59] in short-term (24 h) Cd-treated MCF-7 cells (10 and 100 µM concentrations) and was associated with the role of this molecular chaperone in preventing protein degradation under stress conditions.

#### 2.2.6. Expression Levels of Antioxidant System and Inflammatory Markers

In the same paper, Darwish et al. [59] demonstrated the drastic dose-dependent down-regulation of some genes whose products are involved in the protection against oxidative damage, i.e., glutathione S-transferase omega 1, NAD(P)H quinone dehydrogenase 1, superoxide dismutase 1 and 2, and catalase, and the up-regulation of *HO-1,* coding for the antioxidant heme oxigenase 1 enzyme, which is known to be triggered by Cd in MCF-7 cells via the p38 kinase pathway and via Nrf2 [64]. Moreover, they found that Cd determined a significant dose-dependent accumulation of the mRNAs of cyclooxygenase-2, the tumor necrosis factor-α, and interleukin 8 and 10, whose induction is conceivably linked to inflammation and cell damage in various organs. Interestingly, a reduction in the expression of these genes and a concurrent increase in the expression of the genes for metallothionein-1 and the xenobiotic transporters was obtained by the co-exposure of MCF-7 cells with both Cd and fat-soluble vitamins (mainly D, whereas A and E to a lesser degree). This confirmed the cytoprotective effects of the latter which make them a dietary supplement worth investigating for people at high risk of exposure to Cd.

## 3. Molecular Signatures in Cancer Cells of the Gastrointestinal Tract

### 3.1. Molecular Signatures in Gastric Cancer Cells

The search for Cd exposure-associated molecular signatures has been conducted in cell lines isolated from gastric adenocarcinoma, i.e., MKN28 (well-differentiated), SNU638 (poorly differentiated with the mutated p53 gene) and AGS, the latter characterized by gene expressions typical of tumors that likely process from intestinal metaplasia [65,66]. In these cell lines Khoi et al. [67] examined the effects of a 4 h-treatment with 20 µM Cd on the expression levels of the *PLAUR* gene, coding for urokinase-plasminogen activator receptor (uPAR), demonstrating its time-dependent up-regulation. A more detailed investigation on the sole Cd-treated AGS cells brought evidence that *PLAUR* over-expression was to be ascribed to the activation of the ERK1/2-NFkB-AP1 pathway and that uPAR accumulation was a likely mediator of Cd-induced stimulation of cell invasiveness.

### 3.2. Molecular Signatures in Colon Cancer Cells

The search for Cd exposure-associated molecular signatures has been conducted in the poorly differentiated RKO cell line, which bears mutations in *BRAF* and *PIK3CA* oncogenes, and the HT-29 cell line, isolated from a primary colon adenocarcinoma [68,69].

#### 3.2.1. Molecular Signatures in RKO Cells

The subchronic low-dose exposure of RKO cells to Cd (50 µM for 24 h) [70] resulted in the differential expression of twenty genes, most of which belong to the hsp-coding family and are listed in Table 4, that can be considered as potential carcinogenesis-linked genes under the experimental conditions used. Among the other genes shown to undergo expression alterations, some were related to the detoxication of chemical carcinogens (*AKR1C2*), cell survival (*DCD*, *IDH1*), migration-promoting EMT transition (*P4HB*), and colon cancer metastasis (*PGK1*) [71,72,73].

#### 3.2.2. Molecular Signatures in HT-29 Cells

More recently, dealing with the cyclooxygenase inflammatory pathway, a study by Naji et al. [74] using the luciferase reporter assay demonstrated the early (6 h) and time-dependent 100 nM CdCl_2_-driven increase in *COX2* expression at the transcriptional level in HT-29 colon adenocarcinoma cells. It is widely acknowledged that the cyclooxygenase pathway is involved in colorectal malignant progression, and also promotes the migration of tumor cells via the activation of the ROS-p38-COX-2-PGE_2_ and the ROS-Akt pathways. Therefore, the up-regulation of the inducible COX-2 isoform may be considered as one of the mechanisms by which exposure to environmental Cd pollutants maximizes colorectal malignancy in exposed individuals. In addition, in cells treated with 0.05–10 µM CdCl_2_ for 24 h, Iftode et al. [75] reported the suppression of the expression of *DNMT1* and *DNMT3B* genes, coding for DNA methyltransferase-1 and -3β and which act as hypomethylating agents likely responsible for the carcinogenetic properties of Cd.

## 4. Molecular Signatures in Liver Cancer Cells

The search for Cd exposure-associated molecular signatures has been conducted in HepG2 cells, isolated from a differentiated hepatocellular carcinoma.

Fabbri et al. [76] and Urani et al. [77] performed a whole-genome analysis by cDNA microarray after 24 h exposure of this cell line to 2 and 10 µM CdCl_2_, considered to be low human-relevant metal concentrations. In particular, the work of Urani and coworkers has also focused on intracellular zinc displacement by Cd and its molecular consequences, as zinc is an acknowledged second messenger and transcriptional regulator. Their investigation into gene expression profiling was complemented by a miRNA expression analysis, due to the considerable involvement of their altered regulation in the carcinogenetic process. A group of eleven genes, all belonging to the metallothionein-coding family (i.e., *MT1A*, *-B*, *-E*, *-F*, *-G*, *-H*, *-JP*, *-L*, *-M*, *-X* and *-2A*) were found up-regulated after exposure to CdCl_2_ concentrations whereas the down-regulation of 12 miRNAs and the differential-expression of 949 genes was proven in response to the treatment with the sole higher metal concentration. In particular, the down-regulation affected the genes involved in the liver function pathways (e.g., fatty acid/cholesterol metabolism and hemostasis) while the up-regulation concerned the genes involved in inflammation and cancer progression (e.g., cytokine-cytokine receptor interaction-, focal adhesion- and MAPK signaling). A subset of relevant genes was further submitted to validate their differential expression through real time-PCR (Table 5). 

An analysis of the KEGG database indicated that most validated genes (*CAPN2*, *COL1A1*, *ITGA2*, *ITGA3*, *ITGB1*, *JUN* and *LAMB3*) were associated with focal adhesions whose regulation is involved in liver cancer cell invasion and metastasis, in association with the dysregulation of the cytoskeletal component [78]. Interestingly, Urani and coworkers also found the up-regulation of other adherent junction pathway-related genes, i.e., *SNAI1*, *MET*, *TGFBR2*, *RAC* and *CDC42*, thus confirming the facilitating role of Cd in cell motility and metastatization processes. The other validated genes, i.e., *FOS*, *HSPA6* and *GADD45B*, were associated to the MAP kinase pathway, which is known to play a role in hepatocarcinoma cell survival and tumor growth [79]. It is also of note that the two top pathways of the Cd-dependent dysregulated miRNAs were related to focal adhesions and the MAP kinase cascade.

Distinct sets of 330 and 181 genes were found impaired by the authors in [80] after acute and chronic low concentration-CdCl_2_ treatment of HepG2 cells, respectively. The majority of them were involved in detoxification and metabolic processes in the exposure conditions in the former, and in the regulation of various signaling pathways including the inflammatory and insulin responses in the latter. A subset of relevant genes was further submitted to validate their differential expression through real time-PCR (Table 6).

Under acute exposure to the metal compound, as expected for cells undergoing detoxification, the up-regulation of some members of the metallothionein gene family were found, conceivably driven, at least in part, by the down-regulation of *SPINK1* as reported for colon cancer cells [81]. Down-regulation was also observed for ecto-5′-nucleotidase (a.k.a. CD73), the major enzyme responsible for the enzymatic dephosphorylation of the inflammation-promoting extracellular adenosine 5′-monophosphate nucleotide to the immunosuppressive adenosine. Conceivably, such down-regulation might be an indicator of a pro-inflammatory activity [82]. Interestingly, *CYP3A7*, whose product is involved in drug metabolism, was either down- or up-regulated following acute or chronic exposure, respectively, and this might be associated to the Cd cytotoxic insult in the former and the onset of a detoxification reaction by stressed cells in the latter [83,84]. Following chronic treatment, the up-regulation of the extracellular matrix protein nephronectin may complement the down-regulation of *NT5E* and be related to the stimulation of an inflammatory reaction, according to the evidence of protein localization in inflammatory foci in animal models of hepatitis [85]. Other validated differentially-expressed genes were: (i) *DNAJB9*, whose up-regulation may be associated with the inhibition of p53-induced senescence leading to cell mitogenic signalization and transformation; (ii) *ADH4*, whose lowered expression is linked to the stimulation of several cancer related pathways, including ATR, FOXM1, FOXO, MTOR, NOTCH, and the p53 downstream pathway; (iii) *EGR1* coding for an anti-tumorigenic zinc-finger transcription factor; and (iv) *ID1* whose down-regulation may impair the cell redox state by overproduction of ROS [86,87,88,89].

By comparison with MCF-7 breast tumor cells, 48 h-exposure to Cd also up-regulated the expression of *PRMT5* and *EZH2* methyltransferase genes in HepG2 cells, albeit to a lesser extent, as evidenced by the luciferase reporter assays, thereby confirming the impact of the metal on the expression of the two oncogenic epigenetic regulators also in this neoplastic cytotype [63].

## 5. Molecular Signatures in Lung Cancer Cells

The search for Cd exposure-associated molecular signatures has been conducted in the following cell lines isolated from non-small cell lung carcinoma tissues: (i) H460 characterized by mutant *K-ras* and wild-type *p53* [90,91] and its Cd-resistant derivative RH460 established after selection via the exposure of the parental line to increasing Cd concentrations [92]; (ii) H1299 established from a lymph node metastasis of the lung from a patient who had received prior radiation therapy and endowed with a homozygous partial p53 deletion determining the lack of protein expression; and (iii) A549 characterized by mutant *K*-*ras*, wild-type EGFR and the properties of type II alveolar epithelial cells [93].

### 5.1. Molecular Signatures in H460 and RH460 Cells

Kim et al. [94] studied the induction of the multidrug resistance-associated protein 1 (MRP1), coded by the *ABCC1* gene, by Cd treatment in responsive H460 vs. resistant RH460 cells. This protein is a multitasking transporter broadly involved in many aspects of cell biology and pathology spanning from cell survival and differentiation to inflammation and cancer [95]. The acquired Cd resistance of RH460 cells determines the absence of Cd-induced apoptosis and autophagy, occurring in the parental line, due to the lack of glycogen synthase kinase (GSK)-3β phosphorylation at serine residue and the consequent intracellular relocalization of the molecule [96]. Using inhibitors and siRNAs against MRP1 and GSK-3β and overexpressing GSK-3β-HA, Kim et al. [94] revealed the role played by the kinase in the modulation of the expression of the MRP1 molecular signature through both transcriptional regulation and direct interaction with p-Ser GSK3αβ which intervenes in MRP1 stabilization and intracellular redistribution. The obtained data represent a promising tool for the formulation of GSK-3β serine phosphorylation-inducing chemotherapeutics aimed to treat multidrug resistant lung tumors.

### 5.2. Molecular Signatures in H1299 Cells

It is known that various metals, including Cd, interfere with the localization, folding and function of members of the p53 protein family [97]. Within this context, Adámik et al. [98] examined whether CdCl_2_ impaired the function of p63 and p73 as transcription factors. To this purpose, the different p53 family isoforms were co-transfected with p53 family-dependent luciferase reporter vectors (pGL3-MDM2-APP, pGL3-PGM1 and pGL3-BAX) into p53-deficient H1299 cells. The obtained data demonstrated that the p63 and p73 transactivation of some of the p53-dependent promoters was inhibited by exposure to 20–50 mM CdCl_2._ This was also demonstrated in light of the data confirming the impairment of the binding of the factors’ core domains to p53 consensus sequences, as revealed by electrophoretic mobility shift assays. Thus, the sensitivity of p53 family proteins to Cd appears to be conserved and also active in cell-based assays and, conceivably, the resulting modulation of gene signature expression may play a central role in metal carcinogenesis.

### 5.3. Molecular Signatures in A549 Cells 

Fujiki et al. [99] submitted A549 lung tumor cells to prolonged 20 mM CdCl_2_ exposure and observed the onset of a high proliferative rate, EMT, stress fiber formation, cell locomotion, and resistance to antitumor drugs. From a molecular point of view, the involvement of Notch1 signalization in the Cd-promoted malignant progression was demonstrated, and which was maintained for a considerable time after the removal of CdCl_2_ from the culture medium. In particular, the cell treatment was associated with the up-regulation of *NOTCH1*, *JAG2,* coding for Notch-ligand Jagged-2 protein, and *MMP2*, coding for matrix metalloprotease 2, an invasion-facilitating collagenase which is a prognostic factor for non-small cell lung cancer [100]. On the other hand, the down-regulation of *NOTCH3* was a further gene signature of cell exposure to CdCl_2_. The cumulative results obtained indicated that the transcriptional activity of Notch1 was stimulated by hypoxia-inducible factor 1 (HIF-1) and that the insulin-like growth factor 1 receptor (IGF-1R)/Akt/ERK/p70 S6 kinase 1 (S6K1) cascade could cooperate with Notch1 signaling and HIF-1 following the CdCl_2_ treatment of A549 cells.

## 6. Molecular Signatures in Nasal Septum and Nasopharyngeal Cancer Cells

Since inhalation is the primary route of Cd intake, mainly by cigarette smoking and also by occupational exposure during working activities related to nickel–cadmium batteries, electroplating, and paint pigments, RPMI-2650, CNE-1 and CNE-2 cell lines were used as an in vitro model system to examine the molecular targets of Cd-induced cancer progression in nasal and nasopharyngeal epithelia.

### 6.1. Molecular Signatures in RPMI-2650 Cells

The RPMI-2650 line was isolated from the pleural effusion of a patient with an anaplastic squamous carcinoma of the nasal septum [101]. In the light of the observed Cd-dependent up-regulation of the intracellular ROS level and the down-regulation of cell proliferation, Lee et al. [102] compared the mRNA expression patterns of RPMI-2650 cells grown in control conditions or exposed to 0.75 µM Cd acetate for 72 h via differential display analysis. Following a preliminary analysis of gene expression, *AKR1C3*, coding for the aldo-keto reductase family 1 member C3 protein, was proven to undergo an increase in expression, regulated by Cd at the transcription/translation level; this up-regulation was also confirmed by Western blot analysis. This molecular signature is a hormone activity regulator and prostaglandin F synthase is responsible for monitoring the occupancy of hormone receptors and controlling cell proliferation and differentiation in a hormone-independent way [103]. Based on the reported Cd-induced accumulation of the Nrf2 transcription factor in the nucleoplasm and the restraining of the Cd-dependent increase in the AKR1C3 protein levels by a phosphoinositide 3-kinase (PI3K) inhibitor, it was suggested that the up-regulation of AKR1C3 may result from the augmentation of intracellular ROS, at least in part through the activation of Nrf2 and the onset of PI3K-related signalization, thereby contributing to an adaptive intracellular response to Cd cytotoxicity.

### 6.2. Molecular Signatures in CNE-1 and CNE-2 Cells

CNE-1 and CNE-2 cell lines were established from a well-differentiated and a poorly differentiated nasopharyngeal squamous carcinoma, respectively, a highly invasive and metastatic malignant tumor with unique ethnic and geographic distribution and prominent incidences in South China and some African areas only [104,105]. With the aim to mimic chronic low-level Cd exposure, Peng et al. [106] exposed both cell lines to a non-toxic CdCl_2_ concentration (1 µM) for up to two weeks and observed the acquisition of more proliferative and aggressive characteristics by the cells both in vitro and in vivo. Molecular analyses revealed that chronic Cd treatment induced a remarkable up-regulation of *CCND1*, *CCNE1*, *MYC* and *JUN*, thereby demonstrating the activation of Wnt/β-catenin signaling, which is also in the parallel confirmatory data on increased β-catenin protein immunostaining in Western blots. Further studies highlighted that chronic Cd exposure induced the down-regulation of *CSNK1A1*, coding for the α isoform of casein kinase I, via the hypermethylation of the promoter CpG islands. On the other hand, given that this enzyme is a negative regulator of the Wnt/β-catenin pathway [107], this molecular event might be involved, at least in part, in the exacerbation of the malignant phenotype by neoplastic nasopharyngeal cells.

## 7. Conclusions

Malignant growth is a multistep process driven by accumulating genetic alterations that progressively lead the cells to acquire novel abilities such as self-sufficiency in growth signals, insensitivity to growth-inhibitory signals, evasion of apoptosis, limitless replicative potential, sustained angiogenesis, tissue-invading and metastasizing attitude, oncopromoting inflammatory ability, energy metabolism reprogramming and immunoevasion (the “hallmarks of cancer “ [108,109]). 

Since the establishment of the HeLa cell line in 1951, human tumor cells in culture have represented the most extensively used model system to study cancer biology providing the wealth of information that we currently possess about the molecular, biochemical, and genetic aspects of oncodevelopment and the modes of action of potentially anti-cancer compounds. On the other hand, the clinical relevance and usefulness of these in vitro models is still controversial, mostly due to the lack of interactions with other cell types and the signaling and structural molecule-containing extracellular microenvironments which influence the action of drugs in vivo. Nevertheless, the growing body of data on the molecular profiles and biological characterizations arising from established and novel in vitro preclinical models cannot be ignored, thereby representing a highly useful resource for the testing of scientifical hypotheses focused on the elucidation of carcinogenetic mechanisms and the initial assessment of cancer treatments [110,111].

It is generally accepted that the knowledge of the repertoire of genes transcribed under a given circumstance, such as the exposure to environmental pollutants, may aid in the dissection of the mode of action of the toxicants under investigation [112]. This review has pointed out that different neoplastic cytotypes undergo a wide range of Cd-induced changes in their transcriptional profiles which can consequently influence protein expression, signal transduction, and cell metabolism, and ultimately tie in with the previously listed traits of the carcinogenetic process. Figure 1 summarizes in a Venn diagram the representation the gene signatures discussed, showing the areas of overlap between the different cancer cell models, which is a useful procedure to select the most promising candidates for further translation in in vivo studies.

The individuation of single or panels of novel expression markers can provide a broader picture of the numerous and complex effects exerted by the metal and, in addition, has potential as alternatives to traditional biomarkers for the efficient assessment of the tumor progression stages in diverse cancers. On the other hand, descriptive epidemiological studies can certainly take advantage from detailed studies at the molecular level, which provide new avenues for analytic research. The Cd levels in blood, urine, feces, liver, kidney, hair, and other tissues can be easily measured and can provide an indication of recent or total exposure to Cd, urine being the first choice medium for biomonitoring [113] and inductively coupled plasma-mass spectrometry of hair is a powerful tool to recapitulate the history of Cd exposure [114]. Thus, a combination of a chemical evaluation of Cd with the analysis of the arrays of the so-called “biomarkers of effects”, such as the molecular signatures, can provide a useful tool for the identification of marker genes in risk assessment [115,116], serving in both prognostic and predictive applications for the screening of exposed populations, as already reported in the cases of Cd-induced neuro- and nefrotoxicity [117,118,119].

## Figures and Tables

**Figure 1 cancers-13-02823-f001:**
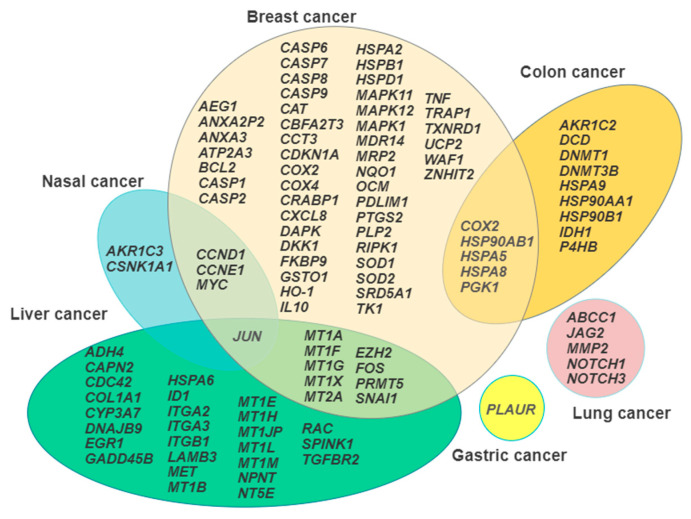
Venn diagram illustrating the gene signature arrays in each cancer cell model examined and the overlapping areas among the model systems.

**Table 1 cancers-13-02823-t001:** Gene expression changes in MDA-MB231 breast cancer cells exposed to 5mM CdCl_2_ for 96 h.

Gene	Protein Product	Up↑ Down↓	Fold Changes
*HSPA5*	Endoplasmic reticulum chaperone BiP	↓	54.2
*HSPA8*	Heat shock cognate 71 kDa protein	↓	4.9
*HSPB1*	Heat shock protein β-1	↑	8.7
*HSPD1*	60 kDa heat shock protein, mitochondrial	↓	2
*HSP90AB1*	Heat shock protein HSP 90-β	↓	2.57
*TRAP1*	Heat shock protein 75 kDa, mitochondrial	↓	9.5
*MT1A*	Metallothionein-1A	↑	2.34
*MT1F*	Metallothionein-1F	↑	3.65
*MT1G*	Metallothionein-1G	↓	18.8
*COX2*	Cytochrome c oxidase subunit 2	↓	3
*COX4*	Cytochrome c oxidase subunit 4	↓	1.9
*BCL2*	Bcl-2	↓	53
*WAF1*	Cyclin-dependent kinase inhibitor 1	↑	10.4
*DAPK*	Death-associated protein kinase-1	↑	55
*RIPK1*	Receptor-interacting protein 1	↑	undetectable in control
*CASP1*	Caspase-1	↑	106
*CASP2*	Caspase-2	↑	3
*CASP6*	Caspase-6	↑	31.3
*CASP7*	Caspase-7	↑	15
*CASP8*	Caspase-8	↑	9.25
*CASP9*	Caspase-9	↑	4.7
*MAPK14*	Mitogen-activated protein kinase p38 α	↓	8
*MAPK11*	Mitogen-activated protein kinase p38 β	↓	4
*MAPK12*	Mitogen-activated protein kinase p38 γ	↑	7
*AEG1*	Astrocyte elevated gene-1 protein	↓	8.5
*PLP2*	Proteolipid protein 2	↑	2
*FOS*	Proto-oncogene c-Fos	↓	3.2
*JUN*	Proto-oncogene c-Jun	↓	3.5

**Table 2 cancers-13-02823-t002:** Breast carcinogenesis-associated gene signatures in MCF-7 cells submitted to chronic CdCl_2_ exposure [53].

Gene	Protein Product	Up↑ Down↓
*ANXA3*	Annexin A3	**↑**
*CCNE1*	Cyclin E1	**↑**
*CRABP1*	Cellular retinoic acid-binding protein	**↑**
*DKK1*	Dickkopf-related protein 1	**↑**
*MT2A*	Metallothionein 2A	**↑**
*PDLIM1*	PDZ and LIM domain protein 1	**↑**
*SRD5A1*	3-oxo-5-alpha-steroid 4-dehydrogenase 1	**↑**
*UCP2*	Mitochondrial uncoupling protein 2	**↑**
*PGK1*	Phosphoglycerate kinase 1	↓
*TK1*	Thymidine kinase, cytosolic	↓

**Table 3 cancers-13-02823-t003:** Metal ion binding-associated gene signatures in the MCF-7 cells submitted to chronic CdCl_2_ exposure [53].

Gene	Protein Product	Up↑ Down↓
*ANXA3*	Annexin A3	**↑**
*ANXA2P2*	Annexin A2 pseudogene 2	**↑**
*ATP2A3*	ATPase, Ca^2+^ transporting, ubiquitous	**↑**
*CBFA2T3*	CBFA2/RUNX1 partner transcriptional co-repressor 3	**↑**
*FKBP9*	FK506-binding protein 9	**↑**
*MT1F*	Metallothionein 1F	**↑**
*MT1X*	Metallothionein 1X	**↑**
*MT2A*	Metallothionein 2A	**↑**
*PDLIM1*	PDZ and LIM domain protein 1	**↑**
*ZNHIT2*	Zinc finger HIT domain-containing protein 2	**↑**
*OCM*	Oncomodulin	↓

**Table 4 cancers-13-02823-t004:** Differentially expressed hsp-coding genes in RKO colon carcinoma cells submitted to subchronic low-dose CdCl_2_ exposure [70].

Gene	Protein Product
*HSP90AA1*	Heat shock protein 90 kDa α (cytosolic), class A member 1
*HSP90AB1*	Heat shock protein 90 kDa α (cytosolic), class B member 1
*HSP90B1*	Heat shock protein 90 kDa β (Grp94), member 1
*HSPA5*	Endoplasmic reticulum chaperone BiP
*HSPA8*	Heat shock cognate 71 kDa protein
*HSPA9*	Heat shock 70 kDa protein 9 (mortalin)

**Table 5 cancers-13-02823-t005:** PCR-validated up-regulated genes in HepG2 hepatocarcinoma cells exposed to 10 µM CdCl_2_ exposure [76,77].

Gene	Protein Product
*CAPN2*	Calpain 2
*COL1A1*	Collagen type I alpha 1 chain
*FOS*	FOS proto-oncogene, AP-1 transcription factor subunit
*GADD45B*	Growth arrest and DNA damage inducible, beta
*HSPA6*	Heat shock 70 kDa protein 6
*ITGA2*	Integrin subunit alpha 2
*ITGA3*	Integrin subunit alpha 3
*ITGB1*	Integrin subunit beta 1
*JUN*	Jun proto-oncogene, AP-1 transcription factor subunit
*LAMB3*	Laminin subunit beta 3

**Table 6 cancers-13-02823-t006:** PCR-validated differentially expressed genes in HepG2 hepatocarcinoma cells exposed to 0.5 µM acute or 0.1 µM chronic CdCl_2_ treatment [80].

Gene	Protein Product	Up↑ Down↓
Acute Treatment	Metallothionein 1F	**↑**
*MT1F*		
*MT1G*	Metallothionein 1G	**↑**
*MT1M*	Metallothionein 1M	**↑**
*CYP3A7*	Cytochrome P450 family 3 subfamily A member 7	**↓**
*NT5E*	Ecto-5′-nucleotidase	**↓**
*SPINK1*	Serine peptidase inhibitor Kazal type 1	**↓**
Chronic Treatment	Cytochrome P450 family 3 subfamily A member 7	**↑**
*CYP3A7*		
*DNAJB9*	DnaJ homolog subfamily B member 9	**↑**
*NPNT*	Nephronectin	**↑**
*ADH4*	Alcohol dehydrogenase 4	↓
*EGR1*	Early growth response protein 1	↓
*ID1*	DNA-Binding protein inhibitor ID-1	↓

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
