# Peer review of "Cadmium-Associated Molecular Signatures in Cancer Cell Models"

_cancers, 2021, doi:10.3390/cancers13112823_

Round 1

Reviewer 1 Report

This is an interesting study, generally well written and structured. The relevant and recent literature are reported on search for Cd exposure-associated molecular signatures in neoplastic cells. In my opinion,  this is a good paper.

Author Response

The Author thanks the Referee for the positive comment on the submitted manuscript

Reviewer 2 Report

This is a very interesting review on the molecular signature of Cd exposure at the level of gene expression changes which provide information on the types of cellular stresses triggered by Cd, as well as on the defense mechanisms developed by cells to counteract the deleterious effects of Cd exposure. Although the review has addressed only human cancer cell lines, the study might provide useful molecular coordinates for future clinical translation. Are there any studies on human professional exposure to Cd? Although experimental animals cannot replicate human exposure, are there any transcriptomic data obtained in tumor-bearing animals exposed to Cd?

The review is very well written. Besides reviewing the literature, the review comments on the signaling pathways and related biological processes, and this is definitely making the list of genes more comprehensive from biological and medical point of view.

There are some points that should be improved:

  1. To make a Venn diagram or other suggestive graphical representation of the genes that are common or distinctive for the presented types of tumor cells. In another diagram you should cluster genes according to the biological processes they are regulating.
  2. If possible, to select the most promising candidate genes for further translation in human studies.
  3. Please define whenever possible the involvement of particular genes in carcinogenesis, or tumor progression or metastasis on one hand, or their role in counteracting the deleterious effects of Cd exposure.
  4. The limitations of the study should be mentioned: are human tumor cell lines reliable for translating results towards human studies?
  5. How are you defining the intrinsic and acquired resistance of cells to Cd?
  6. Please provide for each study the concentration of Cd and the duration of cell exposure.
  7. In my opinion you should introduce reference numbers in the tables.
  8. Although the article is very well written, shorter sentences can make it more easily readable.

Reviewer 3 Report

This manuscript reviews cadmium-associated molecular signatures in cancer cell lines. The author provides a short introduction on cadmium sources and exposure, followed by paragraphs on cadmium uptake pathways, interaction of cadmium with specific molecules, and mechanisms of cadmium-induced molecular damage. The introduction ends on general aspects of molecular reprogramming. The review then largely focuses on molecular signatures detected in breast, gastric, colon, liver, lung and nasopharyngeal tumor cell models that may be transferable to clinical research.

General: The rationale for the concept and structure of the review needs to be clarified. It would be helpful to consider basic concepts on cancer development, the role of cancer cell lines as models of human cancer, and proven or likely target organs of cadmium-induced cancer.

Specific:

  1. The paragraph on uptake pathways for cadmium and cadmium-induced molecular disruption is inadequate. Cadmium uptake by ZIP proteins is only one (minor) uptake pathway. In the body, cadmium is mostly complexed to organic molecules and largely taken up as a complex by a variety of uptake pathways (see for instance doi: 10.1007/s10534-019-00176-6 for an overview).
  2. It is surprising that highly influential reviews on cancer are not discussed at all in the context of the molecular signatures described here (doi: 10.1016/s0092-8674(00)81683-9 and doi: 10.1016/j.cell.2011.02.013). Other relevant concepts, such as “multiple-hit” and “multistage” carcinogenesis are also not covered.
  3. To describe the impact of cadmium on molecular structures and signaling, the highly influential work of Wolfgang Maret, David Petering, Douglas M. Templeton and Jean-Marc Moulis should also be discussed or, at least, cited.
  4. It is unclear why the relevance of cancer cell lines for human cancer is not worth a discussion (see for instance the work M.M. Gottesman, e.g. doi: 10.1093/jnci/djt007).
  5. Reference 1 clearly describes proven or highly likely human cancers, i.e. lung, kidney and prostate. Other likely cancers are breast, endometrial and bladder cancer, and molecular signatures are available in cell models of all these cancers. However, apart from lung and breast cancer, the author reviews rather unusual or rare cancer cell models of gastric, colon, liver and nasopharyngeal cancer, which are unlikely caused by cadmium in humans. The rationale for this selection is unclear. A rational approach would be to select tissues, which store cadmium (see basic concepts of carcinogenesis mentioned in comment 2).

Round 2

Reviewer 2 Report

The authors responded properly to all the raised issues. The clarity of the manuscript was improved.

Reviewer 3 Report

This manuscript reviews cadmium-associated molecular signatures in cancer cell lines. The author provides a short introduction on cadmium sources and exposure, followed by paragraphs on cadmium uptake pathways, interaction of cadmium with specific molecules, and mechanisms of cadmium-induced molecular damage. The introduction ends on general aspects of molecular reprogramming. The review then largely focuses on molecular signatures detected in breast, gastric, colon, liver, lung and nasopharyngeal tumor cell models that may be transferable to clinical research.

General: The author needs to consider the cadmium-associated molecular signatures in human renal and prostate cancer cell lines.

Specific:

Ad 1. Revision satisfactory.

Ad 2. Revision satisfactory.

Ad 3. Revision satisfactory.

Ad 4. Revision satisfactory.

Ad 5. Revision not satisfactory. As mentioned in the reviewer’s previous comments, a rationale for selections of cancer cell lines is to choose cell lines based on tissues, which store cadmium. Key cancer tissues that have NOT been reviewed by the author are kidney, and prostate. Cancers of the prostate and kidneys are 1) common cancers that develop independently of cadmium exposure; 2) these tissues store cadmium independently of other mechanisms of carcinogenesis. Consequently, this combination of occurrences exactly matches the statement of the author: “exposure of cancer cells to Cd compounds may be associated with the development of more malignant phenotypes” and that the plan is to “recapitulate selected studies that report the detection of molecular signatures, as specifically demonstrated by individual gene or whole genome expression profiling”.

Hence, it is mandatory to review molecular signatures of human cancer cell lines exposed to cadmium and that develop more malignant phenotypes when they are exposed to cadmium. For kidney, A-498, ACHN and Caki-1 and -2 are common, but check also “human renal/kidney cancer cell lines”. For prostate LNCaP, PC3 and DU145 cells are common cell lines, but check also for “human prostate cancer cell lines”.

What needs to be more strongly emphasized in this review is the secondary induction of multidrug resistance subsequent to initial development of malignancy by induction of ABC transporters, such as ABCB1. Cadmium is namely a strong inducer of ABCB1 and plays a major role in the development of more malignant phenotypes. This issue should be highlighted more in the review.